# Care by Midwives, Obstetricians, and Dietitians for Pregnant Women Following a Strict Plant-Based Diet: A Cross-Sectional Study

**DOI:** 10.3390/nu13072394

**Published:** 2021-07-13

**Authors:** Deidre Meulenbroeks, Isabel Versmissen, Nanique Prins, Daisy Jonkers, Jessica Gubbels, Hubertina Scheepers

**Affiliations:** 1Máxima Medical Centre, Department of Obstetrics & Gynaecology, 5500 MB Veldhoven, The Netherlands; 2Maastricht University Medical Centre, Department of Obstetrics & Gynaecology, GROW School of Oncology and Developmental Biology, P. Debyelaan 25, 6202 AZ Maastricht, The Netherlands; i.versmissen@student.maastrichtuniversity.nl (I.V.); nm.prins@student.maastrichtuniversity.nl (N.P.); hcj.scheepers@mumc.nl (H.S.); 3Department of Health Promotion, NUTRIM School of Nutrition and Translational Research in Metabolism, Maastricht University, 6200 MD Maastricht, The Netherlands; d.jonkers@maastrichtuniversity.nl; 4Department Internal Medicine, Division Gastroenterology-Hepatology, NUTRIM School of Nutrition and Translational Research in Metabolism, Maastricht University, 6200 MD Maastricht, The Netherlands; jessica.gubbels@maastrichtuniversity.nl

**Keywords:** plant-based, vegan, obstetric care, diet, pregnancy, lactation dietitian, midwife, obstetrician, counseling

## Abstract

With an growing number of people on a strict plant-based diet, its potential effect on pregnancy and lactation becomes increasingly important. It is, however, unclear how obstetric caregivers currently handle and think about a strict plant-based diet in pregnancy. The aim of the study was therefore to evaluate the self-reported knowledge and advice given by Dutch obstetric caregivers and dietitians when treating pregnant women on a strict plant-based diet. A cross-sectional study was performed by sending an online survey to Dutch midwife practices, obstetricians, and dietitian practices. Descriptive statistics are reported. A total of 121 midwives, 179 obstetricians, and 111 dietitians participated in this study. The majority of midwives (80.2%) and obstetricians (93.9%) considered a strict plant-based diet to be a significant risk factor for nutrient deficiency during pregnancy. Maternal dietary preferences, including a potential strict plant-based diet, were discussed at the first prenatal appointment by 59.5% of midwives and 24.1% of obstetricians. A self-reported lack of knowledge concerning the strict plant-based diet was mentioned by 66.1% of midwives and 75.4% of obstetricians. Obstetric caregivers mostly considered the identification of this dietary habit and subsequent referral to a dietitian or a reliable website as optimal care for pregnant women on the strict plant-based diet. However, only 38.7% of dietitians indicated to have sufficient knowledge to counsel these women. Although obstetric caregivers thought that a strict plant-based diet in pregnancy may lead to increased risks of nutritional deficiencies, the majority report to have insufficient knowledge to provide adequate advice. Only a minority referred these women to dietitians, of whom a minority indicated to have adequate knowledge on this specific diet. These results suggest that current care is suboptimal for an increasing number of pregnant women. Women on a strict plant-based diet could benefit from increased knowledge about this topic among obstetric caregivers and dietitians, as well as from clear guidelines regarding this diet during pregnancy.

## 1. Introduction

The popularity of a strict plant-based diet, defined as a diet that consists of fruits, vegetables, grains, legumes, nuts, seeds, herbs, and spices, and excludes all animal products (often also referred to as a vegan diet), is increasing in the western world. In the Netherlands, the estimated number of vegans is about 150,000 in 2020, which is equivalent to 0.86% of the Dutch population [1]. In Great Britain, around 1.16% of the population is vegan, and numbers quadrupled over the last five years [2]. Reports from various western countries show that 75% of all vegans are women, and that 76–81% of all vegans are within the fertile age [3,4,5]. Additionally, research has shown that, in general, women maintain their habitual dietary pattern during pregnancy [6]. Although specific figures for this group are lacking, the number of women on a strict plant-based diet during pregnancy and lactation has likely increased, and will continue to rise in the nearby future. 

Underlying reasons to follow a strict plant-based diet include mainly ethical considerations, environmental concerns, and the possible health benefits [7,8]. People on a strict plant-based diet have been reported to have a lower body mass index, reduced cholesterol and glucose levels compared to omnivores, and might have a reduced risk of developing cardiovascular disease and cancer [9,10]. Additionally, a plant-based diet has a beneficial impact on the intestinal microbiome, with an increase in saccharolytic fermentation and the production of beneficial short chain fatty acids [11]. On the other hand, a strict plant-based diet is associated with a low intake of several micronutrients such as vitamin B12, vitamin D, iodine, calcium, and selenium [12,13]. Maternal prenatal nutrition not only affects maternal health, but is a crucial factor in an unborn child’s neurodevelopment and lifelong mental health as well [14]. Nutritional deficiencies could possibly lead to serious conditions for both mother and child, such as fetal neural tube defects or neurological impairment of the mother [15,16]. 

The Academy of Nutrition and Dietetics states that a well-planned plant-based diet, adjusted for the possible risks of deficiencies is suitable during all stages of life, including pregnancy and lactation [7]. However, the German Nutritional institution advises against a strict plant-based diet during pregnancy due to the increased risk of nutritional deficiencies [17]. In many other countries, no national guidelines exist. In the Netherlands, the Dutch Center of Nutrition advises women on a strict plant-based diet to consult a dietitian for guidance during pregnancy [18]. It is, however, unclear how this is implemented in daily practice, and to what extent this is sufficient to counsel pregnant women. 

The aim of this study was to evaluate the self-reported knowledge and advice given by Dutch obstetric caregivers and dietitians when treating pregnant women on a strict plant-based diet.

## 2. Materials and Methods

### 2.1. Study Design and Setting

The current study was a cross-sectional questionnaire-based survey among Dutch midwives and obstetricians, together responsible for obstetric care, and dietitians in the Netherlands. In the Netherlands, midwives provide obstetric care to low risk women (first line care). The main difference to other healthcare systems is that these midwives are independent healthcare providers. In case of medium or high risk, expressed before or during pregnancy or delivery, women are referred to obstetricians in general, or academic hospitals. In general, about 80% of all Dutch women start their care in first line care, but about 50% of them are referred to obstetricians during pregnancy. Therefore, obstetric care can be provided exclusively by a midwife or obstetrician, but can also be provided by a combination of both. In addition, both midwives and obstetricians could refer pregnant women to a dietitian if necessary. 

### 2.2. Participants and Procedures

#### 2.2.1. Midwives

A total of 200 midwifery practices out of over 500 midwifery practices in the Netherlands (≈40%) were invited to participate in the current study in the first half of 2019 [19]. The approached practices were geographically distributed evenly across the country, taking population density of different areas into account. Midwives were contacted by phone to complete the questionnaire. On request, an e-mail was sent with a link to the online questionnaire to be completed within 4 weeks. 

#### 2.2.2. Obstetricians

In the first half of 2019, The Dutch Society of Obstetrics and Gynecology sent an email invitation by email to all 1453 obstetricians and residents (from here on collectively referred to as obstetricians) in the Netherlands to fill in the online questionnaire within 4 weeks.

#### 2.2.3. Dietitians

The questionnaire for dietitians was posted twice on the Facebook page of the Dutch organization for Dietitians. In addition, 120 randomly selected dietitian practices, geographically distributed evenly across the country, were contacted by email with an invitation link to fill in the online questionnaire within 4 weeks. 

### 2.3. Questionnaire

A validated questionnaire to investigate knowledge concerning the plant-based diet and pregnancy was not available. Therefore, focus group interviews with 20 Dutch women on a strict plant-based who were pregnant or who recently gave birth were used to develop a questionnaire (unpublished work). They, for example, indicated that they often received a negative attitude from obstetric care providers concerning their strict plant-based diet, and they thought it was difficult to get reliable information about their diet from these care providers and from dietitians. The questionnaire for this study was developed based on the topics discussed in the focus groups (Appendix A) and was pretested with six potential survey recipients. 

The questionnaire for midwives and obstetricians took approximately 5–10 min to fill in, and consisted of 19 multiple choice and open questions, including their age, gender, whether they ask pregnant women about dietary preferences, what they advise women on a strict plant-based diet, if they had any nutritional education, and who should be responsible for giving nutritional advice. The questionnaire for dietitians also took approximately 5 min to complete, and consisted of 13 multiple choice and open questions. Participants were asked, for example, about their education on the strict plant-based diet during pregnancy, and if they felt competent to counsel these women. Online questionnaires were conducted via the online survey website SurveyMonkey (https://nl.surveymonkey.com, accessed on 1 February 2019). 

### 2.4. Statistics

The results of the questionnaires were exported to IBM SPSS Statistics 25.0 (IBM Corp., Armonk, NY, USA). The results of midwives, obstetricians, and dietitians were analyzed separately. Categorical questions were analyzed by descriptive statistics using frequencies and percentages. The answers to open questions were categorized and then presented as percentages. Valid percentages are presented. Continuous variables were presented as median with range, or mean with standard deviation, depending on the normality of the distribution.

### 2.5. Details of Ethics Approval and Consent

The local ethical board of the Maastricht University Medical Centre+ waived the project from full ethical review, in line with the Dutch law for medical research with humans (WMO).

## 3. Results

### 3.1. Midwives and Obstetricians

The response rate was 60.5% (*n* = 121) for midwives and 68.4% (*n* = 54) for obstetricians of Dutch hospitals, resulting in a total of 179 responding obstetricians. Background characteristics of the respondents can be found in Table 1. 

The information reported by midwives and obstetricians can be found in Table 2 and Table 3, respectively. At the first prenatal appointment, 59.5% of midwives and 24.1% of obstetricians asked most of the time or always about clients’ dietary preferences. In total, 39.7% and 29.5% of midwives and obstetricians, respectively, stated that a pregnant woman on a strict plant-based diet consulted them in the past year.

In total, 77.7% of the midwives considered themselves to be responsible for advising pregnant women on a strict plant-based diet on nutrition and lifestyle. However, a small group of midwives (12.4%) stated that it is also the responsibility of the patient themselves to find information on this topic. For the obstetricians, the majority (74.3%) believed that the obstetrician should advise their pregnant clients on nutrition and lifestyle. However, almost half of the obstetricians indicated that also the patient herself and the Dutch National Nutrition Center (45.8% and 45.8%, respectively) should provide this information. In addition, the majority of the midwives (81.6%) considered themselves responsible for providing information about breastfeeding and plant-based formula to women on a strict plant-based diet. For obstetricians, this percentage was much lower (43.0%). A total of 20.3% of midwives and 41.9% of obstetricians found that dietitians should have a role in providing this information.

Almost all obstetric caregivers (80.2% of midwives and 93.9% of obstetricians) expected pregnant women to have a higher risk of developing nutrient deficiencies during pregnancy when on a strict plant-based diet. Most midwives (87.6%) and obstetricians (61.3%) would apply additional measures for clients on a strict plant-based diet. The most frequently mentioned policies by midwives were providing extra nutritional advice (30.5%) or a referral to a dietitian (24.8%). They would also check hemoglobin levels (24.0%), vitamin B12 levels (20.7%), and general vitamin status (18.2%). The most frequently mentioned policies by obstetricians was checking general vitamin status (23.5%), advising to use supplements (15.1%), and referral to a dietitian (14.5%). During the period of lactation, midwives and obstetricians would advise their clients to take multivitamins supplements, pregnancy vitamins, and/or to consume extra protein or fat (28.9% and 10.8%, respectively). Some midwives (27.3%) and most obstetricians (56.6%) thought that additional foods or supplements should be advised, but they were not able to specify which ones. 

The majority of both midwives (i.e., 66.1%) and obstetricians (i.e., 75.4%) considered their knowledge to be insufficient to advise pregnant women on the plant-based diet in pregnancy. Furthermore, 68.6% of midwives and 93.9% of obstetricians indicated that their education regarding nutrition was insufficient or nonexistent. The majority (86.8% of midwives and 88.6% of obstetricians) wanted to learn more about a strict plant-based diet in pregnancy, preferably through a short online course. 

According to the midwives and obstetricians, optimal care for pregnant women on a strict plant-based diet would entail systematic identification of such women and referring them to a dietitian or a website with relevant and reliable information. In total, 28.9% of midwives and 16.2% of obstetricians agreed with the statement that every midwife or obstetrician themselves should have the knowledge to advise these women.

### 3.2. Dietitians

The information reported by dietitians can be found in Table 4. A total of 111 dietitians completed the questionnaire. Of these, 22.8% saw at least one pregnant woman on a strict plant-based diet in the past year. 

The majority of the dietitians (96.4%) indicated that their formal training did not cover a strict plant-based diet during pregnancy. An additional course on nutrition that covered the strict plant-based diet was followed by 7.2%. Of all dietitians, 38.7% indicated that they had sufficient knowledge to advise women on a strict plant-based diet during pregnancy. In addition, 95.5% of dietitians would like to learn more about this subject by attending lectures on this topic during conferences (33.6%), completing a course on this topic (26.2%), or by guidance of a national protocol (39.3%).

Dietitians almost exclusively (96.4%) indicated that women should receive information about their strict plant-based diet and pregnancy already before conception. Dietitians themselves would seek for information about a strict plant-based diet during pregnancy on the website of the Dutch Centre for Nutrition (79.1%), the Dutch Magazine for Dietitians (NTVD; 45.5%), ask a colleague (58.2%), and/or by using other information sources such as the internet, websites about nutrition, and scientific evidence (46.4%).

## 4. Discussion

To our knowledge, this is the first study to investigate self-reported knowledge and advice given by obstetric caregivers and dietitians, about a strict plant-based diet during pregnancy specifically. Our study shows that the majority of midwives, obstetricians, and dietitians in the Netherlands feel that they have insufficient knowledge to adequately counsel and guide pregnant women on a strict plant-based diet. 

The availability and communication of information and personal control during pregnancy and childbirth is important for patient satisfaction among all pregnant women [20,21]. This satisfaction is thought to have both short-term and long-term effects on women’s health and the relationship with the infant [21]. It is therefore of great importance to provide women on a strict plant-based diet with sufficient information on this topic, to improve patient satisfaction. However, our study showed that the majority of Dutch midwives and obstetricians indicated that they did not have sufficient knowledge on the topic to provide the necessary information. Most obstetricians and midwives were unfamiliar with a strict plant-based diet during pregnancy. This is in line with a study that addresses health professionals’ knowledge regarding vegetarian nutrition [22]. The low prevalence of a strict plant-based diet may contribute to the lack of knowledge of among obstetric caregivers, in addition to lack of education on the topic. 

The majority of midwives, and almost all obstetricians, in the current study thought that there is an increased risk of developing deficiencies in nutrients, vitamins, or minerals in women on a strict plant-based diet during pregnancy. However, only 60.5% of midwives, and 23.6% of obstetricians, asked about diet at the first prenatal appointment. Furthermore, our research showed that the vast majority of obstetricians, midwives, and dietitians were willing to learn more about a strict plant-based diet in pregnancy, by different ways of education. Most obstetricians and midwives preferred having a signaling function and referring these women to a dietitian or reliable website, in line with various guidelines advising to refer these women to a dietitian [7,18,22]. However, our study showed that most dietitians do not feel competent enough to advise these women either. Therefore, the current advice from the Dutch Centre for Nutrition seems to be insufficient to ensure that pregnant women on a strict plant-based diet receive correct nutritional advice. It should become clear who is responsible for signaling and for advising pregnant women on their diet, so that this specific group of women knows where to go with questions. If we want dietitians to be able to advise pregnant women on a strict plant-based diet, their knowledge on this topic needs to be improved.

The results of the current study can be used as a starting point to improve healthcare for this specific group of women. However, the study has a number of limitations that need to be taken into account. Firstly, as a mass e-mail was sent to all obstetricians with the title “questionnaire about pregnancy and veganism”, participation bias might be induced, with obstetricians that were interested in the topic being overrepresented. Nonetheless, our prevalence rates of knowledge about and attention for a strict plant-based diet during pregnancy would be an overestimation. This further stresses the importance of our findings. Secondly, the questionnaire for obstetricians had some multiple-choice questions in the online questionnaire, whereas some questions for the midwives were open-ended, as some were administered by phone. It might be possible that obstetricians would have provided different answers if they had to complete open ended questions as well. Thirdly, the questionnaire for dietitians did not contain a question about their actual care for pregnant women in the last year. In the Netherlands, dietitian practices are accessible for all people with nutritional questions, including pregnant women. Therefore, having knowledge about diet during pregnancy is relevant for all dietitians. Another limitation is that only obstetricians, midwives, and dietitians were approached to participate in this study. Although midwives and obstetricians are responsible for the majority of obstetric care, general practitioners, maternity assistants, and lactation consultants are also involved in pregnancy and lactation care in the Netherlands [23]. Future research should also address the perceptions of these professionals, as well as the perceptions and experiences of these women themselves. It would be interesting to examine dieticians’ opinion regarding a strict plant-based diet. Additionally, it would be helpful if there would be high-quality courses or e-learnings available, as well as clear guidelines regarding a strict plant-based diet in pregnancy based on scientific research that is supported by dietetic associations and nutritional institutions worldwide. At present, published reviews mostly combine results about vegetarian and plant-based diets [15,24]. Thus far, there are only a few studies performed on a strict plant-based diet specifically, while there are distinct differences reported between the two diets in nutritional intake in, for example, calcium, vitamin B12, vitamin D, omega-3/6 fatty acids, and the source of protein [25,26] Further prospective studies are needed to fill current knowledge gaps regarding the effects of a strict plant-based diet on mother and child. Awaiting these data, obstetric caregivers could consider checking the intake of iodine, vitamin B12, vitamin D, and calcium.

## 5. Conclusions

This study evaluated the self-reported knowledge and advice given by Dutch obstetric caregivers and dietitians when counseling pregnant women on a strict plant-based diet. There is clearly a self-reported lack of knowledge about a strict plant-based diet among midwives, obstetricians, and dietitians. Women on this diet could benefit from more knowledge about this topic among obstetric caregivers and dietitians, and clear guidelines on a strict plant-based diet during pregnancy. 

## Figures and Tables

**Table 1 nutrients-13-02394-t001:** Characteristics of responding midwives, obstetricians, and dietitians.

	Midwives (*n* = 121) *n* (%)	Obstetricians (*n* = 179) *n* (%)	Dietitians (*n* = 111) *n* (%)
**Age**			
<30	44 (36.4)	23 (12.8)	56 (50.5)
31–40	34 (28.1)	89 (49.7)	25 (22.5)
41–50	20 (16.5)	44 (24.6)	17 (15.3)
>51	21 (17.3)	21 (11.7)	13 (11.7)
Unknown	2 (1.7)	2 (1.1)	0 (0.0)
**Gender**			
Male	3 (2.5)	31 (17.3)	0 (0.0)
Female	118 (97.5)	145 (81.0)	111 (100.0)
Unknown	0 (0.0)	3 (1.7)	0 (0.0)

**Table 2 nutrients-13-02394-t002:** Opinions of midwives and obstetricians on nutrition and a plant-based diet during pregnancy.

	Midwives (*n* = 121) *n* (%)	Obstetricians (*n* = 179) *n* (%)
**Diet is a topic during first prenatal consultation**		
Always	23 (19.0)	25 (14.0)
Most of the time	49 (40.5)	18 (10.1)
Sometimes	21 (17.4)	46 (25.7)
Almost never	17 (14.0)	82 (45.8)
I do not ask	7 (5.8)	0 (0.0)
*Other*	2 (1.7)	2 (1.1)
*Unknown*	0 (0.0)	6 (3.4)
**Have a protocol on strict plant-based diet during pregnancy**		
Yes	26 (1.5)	5 (2.8)
No	89 (73.6)	146 (81.6)
Unknown	6 (5.0)	28 (15.6)
**Seen pregnant women on strict plant-based diet last year**		
Yes, >10 women	4 (3.3)	2 (1.1)
Yes, 5–10 women	4 (3.3)	10 (5.5)
Yes, 1–5 women	40 (33.1)	41 (22.9)
No	63 (52.1)	65 (36.3)
Unknown	10 (8.3)	61 (34.1)
**Who is responsible for making sure pregnant women on strict plant-based diet get information about their diet and pregnancy? ***		
Obstetrician	4 (3.3)	133 (74.3)
Midwife	94 (77.7)	57 (31.8)
Nurse	0 (0.0)	47 (38.8)
Dietitian	30 (24,8)	84 (46.9)
Dutch nutrition centre	8 (6.6)	82 (45.8)
Government	1 (0.8)	17 (9.5)
General practitioner	6 (5.0)	31 (17.3)
Patients themselves	15 (12.4)	82 (45.8)
**Expect increased risk for developing deficiencies on a plant-based diet**		
Yes	97 (80.2)	168 (93.9)
No	20 (16.5)	7 (3.0)
Unknown	4 (3.3)	4 (2.2)

* Open ended question for midwives, multiple choice for obstetricians.

**Table 3 nutrients-13-02394-t003:** Opinions of midwifes and obstetricians on prenatal care and their education.

	Midwives (*n* = 121) *n* (%)	Obstetricians (*n* = 179) *n* (%)
**Additional care needed for pregnant women on a plant-based diet in addition to regular care**		
Yes	106 (87.6)	112 (61.3)
Refer to a dietitian	30 (24.8)	26 (14.5)
*Giving nutritional advice*	37 (30.5)	21 (11.7)
*Advise against plant-based diet in pregnancy*	1 (0.8)	1 (0.56)
Blood test for vitamin status	22 (18.2)	42 (23.5)
Checking specifically vitamin B12 in blood	25 (20.7)	11 (6.1)
Checking specifically haemoglobin in blood	29 (24.0)	25 (13.9)
Additional blood tests in general	8 (6.6)	5 (2.8)
Giving supplements	24 (19.8)	27 (15.1)
*Not sure/unknown*	8 (6.6)	6 (3.4)
*Extra ultrasound of fetus*	0 (0.0)	2 (1.1)
*Check relevant literature for advice*	0 (0.0)	13 (7.3)
No	15 (12.4)	67 (37.4)
**Who is responsible for recommendations concerning breastfeeding**		
Obstetrician	1 (0.8)	77 (43.0)
Midwife	98 (81.6)	55 (30.7)
Nurse	11 (1.8)	54 (30.2)
Dietitian	23 (20.3)	75 (41.9)
Dutch Centre for Nutrition	17 (14)	57 (31.8)
Government	2 (1.8)	12 (6.7)
General practitioner	3 (2.6)	15 (8.4)
Children’s health clinic	22 (19.3)	62 (34.6)
*Patients themselves*	19 (16.7)	61 (34.1)
*Lactation consultant*	23 (19.3)	44 (24.6)
*Not sure/unknown*	4 (3.3)	7 (3.9)
**I have sufficient knowledge to advise women on strict plant-based diet in pregnancy**		
Yes	29 (23.9)	13 (7.2)
No	80 (66.1)	135 (75.4)
Not sure/unknown	12 (9.9)	31 (17.3)
**I was educated about nutrition**		
Yes, I think I have learned enough	37 (30.6)	10 (5.6)
Yes, but it was insufficient	54 (44.6)	61 (34.1)
No, I did not/barely learned about nutrition	29 (24.0)	107 (59.8)
Unknown	1 (0.8)	1 (0.8)
***Care for pregnant women on strict plant-based diet could be improved by..****		
*Every midwife/obstetrician should have sufficient knowledge*	35 (28.9)	29 (16.2)
*One midwife/obstetrician in each practice should have sufficient knowledge*	3 (2.7)	32 (17.9)
*The midwife/obstetrician should signal and refer to a dietitian*	37 (30.6)	98 (54.7)
*The midwife/obstetrician should signal and refer to a website/flyer*	34 (28.1)	109 (60.9)
*No specific role*	5 (4.1)	4 (2.2)
Other…	11 (9.1)	6 (3.3)
Unknown	5 (4.1)	7 (3.9)

* Midwives could pick one answer, obstetricians could pick multiple answers.

**Table 4 nutrients-13-02394-t004:** Report of dietitians.

	Dietitians (*n* = 111) *n* (%)
**Education on strict plant-based diet in pregnancy**	
Yes	4 (3.6)
No	92 (82.9)
No, however vegetarian diet and pregnancy were included	15 (13.5)
***Counseled a pregnant women on strict plant-based diet last year***	
Yes, >10 times	0 (0.0)
Yes, 5–10 times	3 (2.7)
Yes, 1–5 times	23 (20.1)
No	85 (76.6)
**Sufficient knowledge to advise pregnant women on strict plant-based diet**	
Yes	43 (38.7)
No	40 (36.0)
I do not know	28 (22.7)
**Pregnant women need to receive information about nutrition ***	
Before pregnancy	107 (96.4)
During pregnancy	70 (63.1)
Extra information is unnecessary	2 (1.8)
***Preference on how to learn more about strict plant-based diet in pregnancy***	
A lecture during a conference	36 (33.6)
A special course	28 (26.2)
*Protocol*	42 (39.3)
*I do not want to learn more*	1 (0.9)
*Unknown*	4 (3.6)

* Dietitians could pick multiple answers.

## Data Availability

Data can be obtained by contacting the corresponding author.

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
