# Peer review of "Care by Midwives, Obstetricians, and Dietitians for Pregnant Women Following a Strict Plant-Based Diet: A Cross-Sectional Study"

_nutrients, 2021, doi:10.3390/nu13072394_

Round 1

Reviewer 1 Report

The authors conducted a cross-sectional study to evaluate the self-reported knowledge and advice given by midwives, obstetricians, and dieticians when counseling pregnant women on a strict plant-based diet. The author revealed that many obstetric caregivers and dietitians thought they had insufficient knowledge to advise women on strict plant-based diet in pregnancy, and they wanted to learn more knowledge about it. The topic in this study is important because maternal nutrition affects the organ development and the risk of the postnatal diseases. The results in this study may be useful to consider the initiative to improve the healthcare for pregnant women and their children.

Major points

Introduction:

  1. The introduction is well described.

Materials and Methods:

  1. I think the number of participants (midwives, obstetricians, and dietitians) are small compared to population.

Results:

  1. Questionnaire form was different in some questionnaires between midwives and obstetricians as described in the manuscript: therefore, it makes difficult to interpret the results that who is responsible for providing the information to the pregnant women on strict plant-based diet, and what could improve care for them. These questionnaires are critical to evaluate how is the advice by the Dutch Center of Nutrition implemented in daily practice. I think more detailed analysis is required to evaluate it.

Discussion

  1. What is the novel findings in this study? The results seems to be similar to those in the studies conducted for other nutrition, such as vegetarian nutrition, as the author described in the manuscript. Furthermore, what new initiative to improve care for pregnant women could be proposed from this study? Please discuss them.

Conclusion

  1. The authors suggested that the knowledge of dietitians on this topic needs to be improved. This is one of the main topics in this study, however, this is not included in Conclusion.

Minor points

  1. There are some typos in the manuscript.

For instance,

Line 109: The results off >> The result of

Line 248: child .. Awaiting these data, >> child. Awaiting these data,

Author Response

Dear reviewer,

We thank you kindly for reviewing our manuscript. We have carefully addressed all comments, and the manuscript has been adapted accordingly.

  1. ‘I think the number of participants (midwives, obstetricians, and dietitians) are small compared to population.’

    We agree that it would have been even better to have a higher number of participants.
    However, the Netherlands is a relatively small country and in our opinion a significant percentage of the obstetrical caregivers with respondents from all area’s in the Netherlands, from single practices and group practices, from academic and regional hospitals participated in our study, as we point out in Materials and methods.

  2. ‘Questionnaire form was different in some questionnaires between midwives and obstetricians as described in the manuscript: therefore, it makes difficult to interpret the results that who is responsible for providing the information to the pregnant women on strict plant-based diet, and what could improve care for them. These questionnaires are critical to evaluate how is the advice by the Dutch Center of Nutrition implemented in daily practice. I think more detailed analysis is required to evaluate it.’

    We have used a basic questionnaire for midwives and obstetricians, supplemented with relevant questions with a focus on the specific group of interest. Additionally, we have added extra information about the Dutch obstetrical care system in line 81-89, to clarify the role of each of the involved caregivers.

  3. ‘What is the novel findings in this study? The results seems to be similar to those in the studies conducted for other nutrition, such as vegetarian nutrition, as the author described in the manuscript. Furthermore, what new initiative to improve care for pregnant women could be proposed from this study? Please discuss them.’

    We agree that the results are similar to those of studies conducted for vegetarian diets, as described in the discussion. However, before we performed this study, this was not yet known. Because of the results in our study, new initiatives to improve care for this specific group of pregnant women can be launched. Suggestions can be found in line 266-269.

  4. ‘The authors suggested that the knowledge of dietitians on this topic needs to be improved. This is one of the main topics in this study, however, this is not included in Conclusion’

    The suggestion that women could benefit from knowledge among dietitians about the plant-based diet can be found in line 283-285. We have also added this to the conclusion of the abstract as well in line 35-37.

  5. ‘There are some typos in the manuscript. For instance, Line 109: The results off >> The result of, Line 248: child .. Awaiting these data, >> child. Awaiting these data,’

    We thank the reviewer for pointing this out, and have corrected typos throughout the manuscript.

Reviewer 2 Report

Thank you for the opportunity to review this interesting study.

This study’s research question is one that has not been addressed in other publications that I am aware of. The results suggest that there is a need for additional training of providers in the Netherlands.

Dietitian is spelled inconsistently throughout the piece. In the United States, the correct spelling is dietitian. Does Nutrients have a preference?

Reference style is inconsistent in terms of capitalization of article titles.

There are relatively minor grammar errors throughout which need to be corrected prior to publication.

The first sentence of the abstract does not seem related to the research being described. This sentence focuses on the potential effects of a vegan diet in pregnancy and lactation while the focus of the manuscript is on attitudes and knowledge of practitioners.

Lines 54-60 present the risks of a vegan diet during pregnancy. However, reference 15 (and other reviews) conclude that well-planned vegan diets can be safe in pregnancy. Ideally, a more balance look at vegan diets would be included in this or the following paragraph.

Additional details are needed about the methods used. The authors state that 200/500 midwifery practices were contacted. How were these practices selected? Could the selection procedure have introduced bias? Were the dietitian practices that were randomly selected to be contacted ones that saw pregnant and lactating women?  Were survey questions pretested with potential survey recipients to insure clarity? Were scales used (for example with questions like extent of knowledge about vegan diets – did respondents indicate where on a scale of 1 to 10, say, they would be?)?

Line 107 – tools used such as Survey Monkey and SPSS are not typically included in the reference list.

Lines 120-122 – why are there 2 different response rates for obstetricians?

It would have been interested to have also asked providers about another nutrition-related issue such as iron-deficiency anemia or diabetes to determine if they were lacking in overall nutrition knowledge or in knowledge of a strict plant-based diet.

Table 2 – question about number of women on strict plant-based diet seen in past year includes “Yes, 0-5.” This should probably be 1-5.

Was there a reason that only midwives and physicians were queried about their opinions as to the safety of a vegan diet in pregnancy? Were dietitians also asked about this?

Author Response

Dear reviewer,

We thank you kindly for reviewing our manuscript. We have carefully addressed all comments, and the manuscript has been adapted accordingly.

  1. ‘Dietitian is spelled inconsistently throughout the piece. In the United States, the correct spelling is dietitian. Does Nutrients have a preference?’

    We have changed the spelling to ‘dietitian’ throughout the manuscript, in line with the preference of the journal.

  2. ‘Reference style is inconsistent in terms of capitalization of article titles.’

    We have corrected the reference style.

  3. ‘There are relatively minor grammar errors throughout which need to be corrected prior to publication.’

    We have checked the article for grammar errors and corrected them.

  4. ‘The first sentence of the abstract does not seem related to the research being described. This sentence focuses on the potential effects of a vegan diet in pregnancy and lactation while the focus of the manuscript is on attitudes and knowledge of practitioners.’

    We have changed the introduction of the abstract in line with your comment, line 17-18.

  5. ‘Lines 54-60 present the risks of a vegan diet during pregnancy. However, reference 15 (and other reviews) conclude that well-planned vegan diets can be safe in pregnancy. Ideally, a more balance look at vegan diets would be included in this or the following paragraph.’

    We have included the possible positive health effects of a strict plant-based diet in lines 56-61 and the possible risks in lines 61-67. The Academy of Nutrition and Dietetics states that a well-planned plant-based diet is suitable during pregnancy, when taking the possible risks into account. We have tried to clarify this contradiction in the introduction.

  6. ‘Additional details are needed about the methods used. The authors state that 200/500 midwifery practices were contacted. How were these practices selected? Could the selection procedure have introduced bias? Were the dietitian practices that were randomly selected to be contacted ones that saw pregnant and lactating women?  Were survey questions pretested with potential survey recipients to insure clarity? Were scales used (for example with questions like extent of knowledge about vegan diets – did respondents indicate where on a scale of 1 to 10, say, they would be?)?’

    - The approached midwifery practices were geographically evenly distributed across the Netherlands, taking population density of different areas into account by inviting relatively more midwifery practices from dense areas to participate in this study. This information is now added to the article, line 94-95. This procedure minimized selection bias. We cannot rule out that healthcare providers with special interest in nutrition were more likely to respond. But even when taking this in consideration, the conclusions would be the same.
    - The dietitian practices that were contacted for this study are accessible for all people with nutritional questions, including pregnant women. As mentioned in the results and in Table 2, not all respondents counselled pregnant women on the strict plant-based diet in the past year. Whether the responding dietitians did counsel pregnant women on a regular diet was not part of the survey. We acknowledge this limitation and have added a comment on this in the manuscript, line 256-269.
    - The survey questions were indeed pretested with potential survey recipients. This information is added to the article, line 117.
    - The questionnaire included multiple choice questions and open questions. This information is now added to the article, line 119 and line 123.

  1. ‘Line 107 – tools used such as Survey Monkey and SPSS are not typically included in the reference list.’

    We have removed the references for both tools and corrected this in the article.

  2. ‘Lines 120-122 – why are there 2 different response rates for obstetricians?’

    The first response rate is about midwives, the second response rate is about obstetricians. We have now clarified this in the article, line 142.

  3. ‘It would have been interested to have also asked providers about another nutrition-related issue such as iron-deficiency anemia or diabetes to determine if they were lacking in overall nutrition knowledge or in knowledge of a strict plant-based diet.’
    In the Netherlands, all pregnant women are checked for anemia in the first trimester and around 27 weeks. Midwives and obstetricians have (shared) protocols that also contain information about anemia and nutrition. Additionally, there are shared protocols about (gestational) diabetes. All pregnant women get a glucose test in the first trimester and an additional oral glucose tolerance test at 24-28 weeks will be performed if indicated. When women are diagnosed with (gestational) diabetes, all obstetricians are able to give these women a general nutritional advice, but they will also receive more in-depth information from a dietitian or specialised nurse. As this is part of standard Dutch practice, we assume that knowledge on anemia and diabetes is sufficient among Dutch midwives and obstetricians.
    More specific knowledge about nutrition in general and restrictive diets specifically is not discussed during medical training in the Netherlands, as also mentioned in this study (line 179-180). We assume that knowledge on for example gluten free diet, paleo diet and raw food diet will be lacking, but this was indeed not discussed in the questionnaire.

    It would have indeed been interesting to learn more about this, but this was not the focus of our study.

  1. ‘Table 2 – question about number of women on strict plant-based diet seen in past year includes “Yes, 0-5.” This should probably be 1-5.’

    We have corrected this to 1-5.

  2. ‘Was there a reason that only midwives and physicians were queried about their opinions as to the safety of a vegan diet in pregnancy? Were dietitians also asked about this?’

    Women who participated in our preliminary focus groups repeatedly mentioned that their obstetrical caregivers had a negative attitude towards their plant-based diet during pregnancy. Some of them were told by their obstetrician or midwife to ‘Just go and eat meat’ or ‘You’re a bad mother if you are on this diet’. Their opinion is of importance because they are the primary caretakers for pregnant women and their opinion influences the advice these women get from their caregiver (or a referral to a dietitian). The dietitians, however, will advise women on a strict plant-based diet because of their nutritional knowledge, or they will not because of a knowledge gap. We therefore did not ask dietitians on their opinions on the safety of the strict plant-based diet. In hindsight, this would have been interesting because two dietitians reported in an open field for remarks at the end of the questionnaire that they did not support the strict plant-based diet during pregnancy at all.
    We have added an extra comment about the focus groups in line 110-115. In addition, we have added a comment to the ‘recommendations for future research’ in line 265-266 as well.

Round 2

Reviewer 1 Report

The authors have addressed most of my concerns and the manuscript has been revised well.